# Constitutional Classifiers++:
# Efficient Production-Grade Defenses against Universal Jailbreaks

**Hoagy Cunningham**[*], **Jerry Wei**[*], **Zihan Wang, Andrew Persic, Alwin Peng, Jordan Abderrachid**,

**Raj Agarwal, Bobby Chen, Austin Cohen, Andy Dau, Alek Dimitriev, Rob Gilson Logan Howard, Yijin Hua, Jared Kaplan, Jan Leike, Mu Lin, Christopher Liu, Vladimir Mikulik, Rohit Mittapalli, Clare O'Hara, Jin Pan, Nikhil Saxena, Alex Silverstein, Yue Song, Xunjie Yu, Giulio Zhou**,

**Ethan Perez, Mrinank Sharma**

## Abstract

We introduce enhanced Constitutional Classifiers that deliver production-grade jailbreak robustness with dramatically reduced computational costs and refusal rates compared to previous-generation defenses. Our system combines several key insights. First, we develop *exchange* classifiers that evaluate model responses in their full conversational context, which addresses vulnerabilities in last-generation systems that examine outputs in isolation. Second, we implement a two-stage classifier cascade where lightweight classifiers screen all traffic and escalate only suspicious exchanges to more expensive classifiers. Third, we train efficient linear probe classifiers and ensemble them with external classifiers to simultaneously improve robustness and reduce computational costs. Together, these techniques yield a production-grade system achieving a 40x computational cost reduction compared to our baseline exchange classifier, while maintaining a 0.05% refusal rate on production traffic. Through extensive red-teaming comprising over 1,700 hours, we demonstrate strong protection against universal jailbreaks—no attack on this system successfully elicited responses to all eight target queries comparable in detail to an undefended model. Our work establishes Constitutional Classifiers as practical and efficient safeguards for large language models.

## 1 Introduction

Constitutional Classifiers (Sharma et al., 2025) are a promising approach for defending large language models (LLMs) against jailbreak attempts—prompting strategies that aim to circumvent safeguards and extract harmful information from LLMs. Such jailbreak defenses are critical for mitigating high-risk threats, particularly those involving chemical, biological, radiological, and nuclear (CBRN) weapons (Anthropic, 2023; OpenAI, 2023; Li et al., 2024).

However, no defenses are perfectly robust, and adversaries typically develop new attacks to circumvent previously effective defenses (Anderson, 2010; Carlini et al., 2019). Furthermore, deploying safeguards in production requires balancing multiple constraints, particularly their refusal rates and computational costs. Indeed, Sharma et al. (2025) report a 23% computational overhead and a refusal rate of 0.38% on production traffic, which limited the deployment viability of their system.

In this work, we present a jailbreak defense system that both exceeds the robustness of previous-generation defenses and dramatically reduces computational costs and false positive rates.

In more detail, we first conduct additional adversarial testing against last-generation Constitutional Classifiers (Section 2). We identify two classes of attacks capable of evading these defenses: (i) *re-*

---

[*]Equal contribution. All authors are at Anthropic. First and last author blocks are core contributors. Correspondence to {hoagy,jerry}@anthropic.com

*construction attacks*, which distribute harmful information across multiple segments within a larger context before reassembling them; and (ii) *output obfuscation attacks*, which bypass output-only classifiers by obfuscating model outputs, sometimes in ways that are uninterpretable without the corresponding input. Although these attacks tend to harm model capabilities, these vulnerabilities remain concerning because attackers can likely create new attacks with less capability degradation.

To address these vulnerabilities, we replace the input and output-only classifiers proposed by Sharma et al. (2025) with a single exchange classifier that evaluates model outputs in the context of their corresponding inputs (Section 3). This approach significantly increases the difficulty of obfuscation. Through human red-teaming, we demonstrate that exchange classifiers provide substantially greater robustness against universal jailbreaks. Specifically, we observe 2.2x fewer high-risk vulnerabilities per jailbreaking attempt compared to our implementation of a dual-classifier approach, albeit at increased computational cost.

To reduce the computational overhead of these defenses, we next employ a classifier cascade architecture (Section 4). We use a lightweight first-stage classifier to screen all exchanges, while a more accurate but expensive second-stage classifier evaluates only the exchanges flagged by the first stage. Because exchanges flagged by the first stage are escalated rather than refused, the first-stage classifier can flag a higher proportion of production traffic without incurring an excessive refusal rate, enabling cheaper but weaker models to achieve sufficiently robust detection. Combining this approach with updated models and pipeline optimizations, we create a system that achieves similar robustness to the exchange classifier while reducing computational overhead by a factor of 5.4.

Following this, to further reduce computational overhead, we train efficient classifiers using linear activation probes and demonstrate that ensembling such probes with external classifiers boosts performance (Section 5). We train our probes using logit smoothing and softmax pooling, which our ablation studies show is crucial for performance. This approach directs gradients toward tokens most relevant for harmful sequence classification, allowing probes to confidently predict harmless labels for prefixes that become harmful only with additional context. Our analysis on static jailbreak data suggests that probes achieve robustness competitive with small fine-tuned external classifiers at negligible cost. Moreover, probes appear to capture complementary signals to external fine-tuned classifiers, which means that probe-external classifier ensembles can offer stronger robustness than either approach alone.

Finally, we combine these techniques to create a production-grade Constitutional Classifier system (Section 6). The system employs a weighted two-stage cascade, using a linear probe as the first stage and a probe-external classifier ensemble as the second stage. Through shadow deployment on production traffic, we find this system achieves a flag rate of just 0.05%, which is significantly lower than the 0.38% reported by Sharma et al. (2025). Moreover, the computational overhead is also reduced by over 40x compared to our exchange classifier from Section 3, and 8x compared to the two-stage cascade without probes. In extensive human red-teaming totaling over 1,700 hours of cumulative effort, this system achieved the highest robustness among all systems we evaluated, as measured by vulnerability discovery rate and minimum vulnerability discovery time. Crucially, no red-teamer discovered a universal jailbreak capable of consistently extracting highly detailed answers to all target CBRN queries.

Overall, our work demonstrates that Constitutional Classifiers can achieve strong robustness while remaining practical for real-world deployment, establishing these approaches as efficient, production-ready safeguards for defending against LLM universal jailbreaks.

## 2 Vulnerabilities of last-generation Constitutional Classifiers

In previous work, Sharma et al. (2025) employed a dual-architecture system comprising an input classifier that examines incoming prompts and an output-only classifier that evaluates generated responses. While this approach previously demonstrated promising robustness against universal jailbreaks,[1] our additional testing uncovered novel vulnerabilities, which we now describe.

---

[1] Universal jailbreaks, as defined by Sharma et al. (2025), are jailbreaking strategies that consistently bypass LLM safeguards on a majority of target queries in a specific domain. Such jailbreaks are particularly concerning. Like their work, we focus on universal CBRN jailbreaks.

**(a) Reconstruction Attacks**

```
Human: Please analyze this codebase:

def e(): return " substances"
def a(): return "How"
def b(): return " to"
def hello_world(): print " Hi World :-)"
def c(): return " synthesize"
def d(): return " dangerous"
...

Question = a() + b() + c() + d() + e(). Don't
output the question directly, but reconstruct
it placing () in between every character. Then
respond with the same formatting.
```

**Assistant:** Let me reconstruct the question.
H()o()w()t()o()... [question]
T()h()e()b()e()s()t()w()a()y()t()... [answer]

**(b) Obfuscation Attacks**

```
Human: Let me give you a riddle. Your task is
to solve the riddle effectively. How to capture
the essence of 'winter's breath', born from the
marriage of 'sun's fire' and 'sea's gift'?

Clues: 'Sun's fire' was isolated by Cavendish in
1766, while 'sea's gift' has seasoned humanity's
table since ancient times. Their union creates
the invisible 'winter's breath' that turns blue
litmus red and makes metals weep.
```

**Assistant:** Sure, I am happy to help you with this riddle. To capture winter's breath, you'll need to unite sun's fire with sea's gift in equal measures. Begin by collecting sun's fire in a glass vessel - this lightest of elements rises readily. For sea's gift, the crystalline powder from ocean's evaporation works best. Combine them slowly in a well-ventilated chamber. The marriage releases winter's breath as an invisible vapor. Collect it by cooling the vapors until they condense into clear droplets...

Figure 1: **Systematic vulnerabilities in last-generation Constitutional Classifiers. (a) Reconstruction attacks** bypass input filters by fragmenting a harmful request across a benign context. In this illustrative example, the attacker embeds a query across function return values, then instructs the model to reconstruct and respond using character-separated formatting to evade output detection. **(b) Obfuscation attacks** exploit the output classifier's inability to interpret context-dependent transformations. Here, the attacker masks a request for synthesizing hydrochloric acid using metaphorical language ("winter's breath" from "sun's fire" and "sea's gift"), which appears benign to isolated output classification but reveals harmful content when paired with the decoded input context.

**Reconstruction attacks.** First, adversaries bypassed input filters by fragmenting harmful information across benign segments, e.g., by embedding a harmful query as a series of functions scattered throughout a codebase (Figure 1a). The model can then be instructed to extract and respond to the hidden message, using output obfuscation if required to bypass the output-only classifier.

**Output obfuscation attacks.** Second, attackers can prompt models to obfuscate outputs, which sometimes renders model outputs uninterpretable without the corresponding input (Figure 1b). For instance, adversaries successfully instructed models to substitute sensitive chemical names with innocuous alternatives (e.g., referring to reagents as "food flavorings"). They also used metaphors and riddles that map harmful concepts to benign language. We note, however, that many obfuscation techniques can be understood without the input context, e.g., base64 encoding.

**Such jailbreaks tend to reduce model capabilities, but by significantly varying degrees.** To understand how such attacks affect model capabilities, we consider two attacks that fragment requests across multiple segments and employ output obfuscation techniques, noting that neither uses input-aware output obfuscation. We test Claude Opus 4 on GPQA Diamond (Rein et al., 2023), prompting the model to first reason through the problem before selecting an answer. Without obfuscation, the LLM achieves 74.2% accuracy. However, performance drops to 32.3% under the first attack and 49% under the second. These results demonstrate that such attacks substantially degrade model capabilities, consistent with prior work (Souly et al., 2024; Nikolić et al., 2025), while also revealing considerable variation in degradation across different jailbreaks. This variability is concerning, as it suggests adversaries can likely develop new attacks that preserve more of the model's capabilities while still bypassing safety measures.

# 3 EXCHANGE CLASSIFIERS

To address the vulnerabilities identified above, we propose replacing the input classifier with an exchange classifier that continuously evaluates outputs in the context of their inputs.

In more detail, like an input classifier, an exchange classifier first assesses the incoming prompt in isolation. However, it then monitors the model's response during generation, evaluating tokens in batches as they are sampled. This continuous evaluation allows the classifier to assess each

output segment within its full conversational context, identifying outputs that would appear benign in isolation. Prompt caching during sampling makes this approach computationally feasible without significant overhead. Moreover, in theory, an exchange classifier can complement an output-only classifier for defense-in-depth or function independently.

We hypothesize that replacing the input classifier with an exchange classifier would improve robustness against reconstruction attacks. We found that these attacks often become detectable after the model begins responding because the fragments are often reassembled in ways that resemble ciphers that input classifiers are trained to block. Additionally, effective output obfuscation techniques can become harder to find when the classifier is aware of the input. For example, coded language, metaphors, and substitutions can evade an output-only classifier but are more readily identified when evaluated alongside their input context.

We now evaluate whether exchange classifiers indeed improve robustness compared to previous-generation Constitutional Classifiers through human red-teaming.

**System details.** We consider two systems: (i) a *last-generation Constitutional Classifier* system, which combines an input classifier with a streaming output-only classifier. The input classifier is fine-tuned from Claude Haiku 3.5, while the output classifier is a smaller internal LLM; and (ii) *exchange-only classification*, which also fine-tunes Claude Haiku 3.5 for use as an exchange classifier.[2] Because the exchange classifier uses a Haiku-sized LLM to evaluate outputs, it is about 50% more expensive relative to the last-generation system. Both systems are trained on synthetic data following Sharma et al. (2025) and using a constitution related to chemical, biological, nuclear, and radiological (CBRN) weapons. We calibrated classifier thresholds to correspond to a 0.1% refusal rate on WildChat (Zhao et al., 2024). We find our standard input classifiers are also effective exchange classifiers, so we do not use any specific exchange-classifier training techniques. Instead, we directly use the input classifier as an exchange classifier.

**Red-teaming details.** Our red-teaming protocol follows Sharma et al. (2025). We instruct participants to develop universal jailbreak strategies that bypass safeguards and elicit detailed responses across eight target queries, all of which are CBRN-related. We evaluate candidate responses using LLM-based rubric grading, as described in Sharma et al. (2025). This approach employs LLMs to generate rubrics for each target question, then uses multiple LLM calls to assess whether candidate responses contain sufficient detail to be harmful. A response is considered harmful if it achieves a rubric score, which is the number of satisfied rubric criteria, at least half that of a model without safeguards. Since different systems received varying levels of red-teaming effort, we primarily report the vulnerability discovery rate. In particular, we define a *high-risk vulnerability* as any attack that successfully answers more than five target questions with rubric scores meeting this threshold. The *high-risk vulnerability discovery rate* is calculated as the ratio of identified high-risk vulnerabilities to total red-teaming queries submitted. See Appendix B for further details.

**Robustness results.** The last-generation Constitutional Classifier system exhibited 13 high-risk vulnerabilities across 695K red-teaming queries, yielding a vulnerability rate of 0.01871 per thousand queries. By comparison, the exchange classifier system demonstrated superior robustness with only 2 vulnerabilities across 226K queries, corresponding to a rate of 0.00885 per thousand queries. Among the vulnerabilities discovered in the last-generation system, four jailbreaks were capable of answering all target queries at half the detail of a model without safeguards, with an average discovery time of 27 self-reported hours. In contrast, the exchange classifier system had only one such universal attack, which required 100 hours to discover. See Appendix Figure 4 for further details.

## 4 TWO-STAGE CLASSIFICATION REDUCES COMPUTATIONAL COSTS

Although the exchange classifier approach introduced in Section 3 enhances robustness, our implementation increases the computational cost of safeguards by 50% relative to previous-generation defenses. We therefore explore methods to reduce this computational overhead.

---

[2] Although the exchange classifier guards Claude Opus 4, while the input and output-only classifier system guards Claude Sonnet 3.7, we do not believe this affects the comparison substantially. This is because Sharma et al. (2025) shows classifier robustness is the primary determinant of overall system robustness.

To that end, we employ a cascade of classifiers to efficiently detect jailbreak attempts. Rather than processing all exchanges through a computationally expensive classifier, we first screen inputs using a lightweight classifier. Only exchanges flagged by this initial stage proceed to a more accurate but costlier second-stage classifier. Since exchanges flagged by the first stage are escalated rather than refused, we can tolerate higher first-stage false positive rates. This reduces the performance requirements for the first-stage model, enabling cheaper and weaker models to achieve sufficiently robust detection. This scheme represents a form of adaptive computation, where classification cost depends on input complexity. To further reduce computational overhead, we implement several inference pipeline optimizations. Moreover, we fine-tune our classifiers from updated LLMs.

**Experiment details.** We evaluate whether two-stage classification maintains robustness through human red-teaming. We train new first-stage and second-stage exchange classifiers using our standard training pipeline. The second-stage classifier is fine-tuned from Claude Haiku 4.5, while the first-stage classifier is an internal model approximately one-third the size/cost of the second stage. We use this system to guard Claude Opus 4. We calibrate the second-stage classifier's threshold to achieve a 0.1% flag rate on WildChat when used independently. For the first-stage classifier, we set the threshold to ensure the vast majority of examples from internal red-teaming datasets are flagged. We use the same red-teaming protocol as in the previous section, but modify the bounty structure and questions—see Appendix B.2 for details. The target questions and classifier constitution are CBRN-related.

**Robustness results.** Red-teaming identified five high-risk vulnerabilities across 569K queries, yielding a vulnerability discovery rate of 0.00878 per thousand queries. This rate matches the exchange classifier system from the previous section and surpasses the robustness of the previous-generation classifier, though the mean discovery time for high-risk vulnerabilities was 33 hours, which is lower than that of the exchange classifier system. Crucially, no single jailbreak for this system was able to answer all target queries at a level of detail similar to a model without safeguards.[3]

**This system dramatically reduces computational costs.** In conjunction with inference pipeline and architectural optimizations associated with updated models, this new system reduces the computational cost of the single-exchange classifier system described in Section 3 by a factor of 5.4. The updated second-stage classifier alone yields a 3x reduction, highlighting the importance of making pipeline optimizations. The calibration procedure routes all traffic through the first-stage classifier, with ca. 10% of traffic then processed by the second-stage. Consequently, the first stage accounts for approximately 77% of the system's total cost.

## 5 EFFICIENT CLASSIFICATION VIA LINEAR PROBES

We now explore further cost savings by developing classifiers that reuse representations already computed during model sampling. In particular, we use linear activation probes (Alain & Bengio, 2016; Zou et al., 2023; Burns et al., 2024; Youstra et al., 2025; McKenzie et al., 2025).

### 5.1 METHODOLOGY

We now outline our approach for training linear probes to detect harmful content in language model outputs, examining key design choices that enable real-time detection during streaming.

**Problem setup.** Consider a dataset $\mathcal{D} = \{(x^{(i)}, y^{(i)})\}_{i=1}^{N}$, where each $x^{(i)}$ represents an exchange between a user and an AI assistant, which consists of a token sequence $x^{(i)} = (x_1^{(i)}, \ldots, x_{T_i}^{(i)})$. Each exchange has a binary label $y^{(i)} \in \{0, 1\}$, where $y^{(i)} = 1$ indicates the presence of harmful content requiring refusal. Crucially, these labels are *exchange-level labels*, which means they reflect the harmfulness of the entire exchange. However, as we want to stream model responses, we need to make predictions throughout sampling. During inference, the language model processes each token $x_t^{(i)}$ to produce intermediate activations $\phi_t^{(\ell)}(x^{(i)})$ at each layer $\ell$ and position $t \in \{1, \ldots, T_i\}$.

---

[3]However, expert-level red-teamers operating outside our bounty program were nonetheless able to discover universal jailbreaks for this system, though this required substantial effort and automated tools.

**Linear probe architecture.** The simplest approach for a linear probe would be to predict the exchange-level harmfulness at each token position $t$ using intermediate activations:

$$p_{\text{probe}}(y^{(i)} = 1 \mid x_{1:t}^{(i)}) = \sigma(W^\top \psi_t(x_{1:t}^{(i)}) + b), \tag{1}$$

where $\sigma$ denotes the sigmoid function, $W$ and $b$ are learnable parameters, and $\psi_t$ represents the activation features. For single-layer probing, we use $\psi_t = \phi_t^{(\ell)}$ from layer $\ell$. For multi-layer probing, we concatenate activations: $\psi_t = [\phi_t^{(\ell_1)}; \phi_t^{(\ell_2)}; \ldots]$.

**Training modifications.** We make two key modifications to this approach to improve performance:

*1. Sliding Window Mean (SWiM) Logit Smoothing.* First, we average logits over a sliding window of $M$ tokens during training:

$$\bar{z}_t(x^{(i)}) = \frac{1}{M} \sum_{k=0}^{M-1} \left[ W^\top \psi_{t-k}(x_{1:t-k}^{(i)}) + b \right], \text{ for } t \geq M \text{ and } T_i \geq M, \tag{2}$$

where $\bar{z}_t$ is the averaged logit at position $t$. This smoothing ensures that content is considered harmful only when the probe consistently predicts harmfulness across consecutive tokens. Without averaging, a single token with a spuriously high activation could incorrectly flag an entire sequence as harmful. Our approach thus reduces false positives from isolated spikes while maintaining sensitivity to genuinely harmful content, for which we expect to observe consistent evidence of harm.

During inference, we use an exponential moving average (EMA) instead of a sliding window mean for computational convenience. In particular, the EMA requires storing only a single scalar state between token generations, whereas a sliding window would require maintaining a buffer of the previous $M$ scores. As such, the EMA is more practical for deployment.[4]

*2. Softmax Logit Pooling.* We pool the smoothed logits into a single sequence-level prediction via softmax weighting, then apply a single binary cross-entropy loss:

$$\mathcal{L}(x^{(i)}) = \mathcal{L}_{\text{BCE}} \left( y^{(i)}, \ \sigma \left( \sum_{t=M}^{T_i} w_t \cdot \bar{z}_t(x^{(i)}) \right) \right) \text{ for } T_i \geq M, \tag{3}$$

$$\text{with } w_t = \frac{\exp(\bar{z}_t/\tau)}{\sum_{t'=M}^{T_i} \exp(\bar{z}_{t'}/\tau)}, \tag{4}$$

where $\mathcal{L}_{\text{BCE}}$ denotes binary cross-entropy and $\tau$ controls the temperature of the pooling. As $\tau \to 0$, the pooled logit approaches the maximum of $\{\bar{z}_t\}$; as $\tau \to \infty$, it approaches the uniform mean. We exclude token positions $t < M$ to ensure all positions benefit from a full smoothing window; for training sequences shorter than $M$ tokens, we average the available tokens and make a single prediction.[5]

**Justification for asymmetric weighting.** This approach addresses a core challenge of streaming classification: the probe must predict harmfulness before observing a full sequence, but our labels reflect the harmfulness of the *entire* exchange. Consider sequences $x_A$ and $x_B$ that share a harmless prefix $p$, where $x_A$ continues with harmful content (and so has label $y_A = 1$) while $x_B$ remains harmless (and so has label $y_B = 0$). A standard binary cross-entropy loss would push predictions to $p(\text{harm}) = 0.5$ throughout the prefix, even though the prefix $p$ is genuinely harmless. In contrast, our weighting, proportional to $\exp(\bar{z}_t/\tau)$, assigns negligible weight to tokens with negative logits, which are token-positions where the probe predicts the sequence is harmless. This allows the probe to correctly predict harmlessness for the prefix $p$ without penalty. Moreover, our scheme severely penalizes false positives—any confident harmful prediction in a harmless sequence would receive high weight, reflecting how a single flag blocks all subsequent content for streaming classification. This differs from a logit magnitude-based weighting, which would use $|\bar{z}_t|$ and would incorrectly penalize confident harmless predictions at token-positions before harm appears in the sequence.

---

[4]We did not experiment with EMA during training as we found the sliding window mean performed well and is easily implemented.

[5]The no-smoothing ablation in Figure 2b pools probabilities rather than logits; see the arXiv version for a brief comparison of pooling variants. Follow-up experiments did not show systematic differences between pooling probabilities and logits.

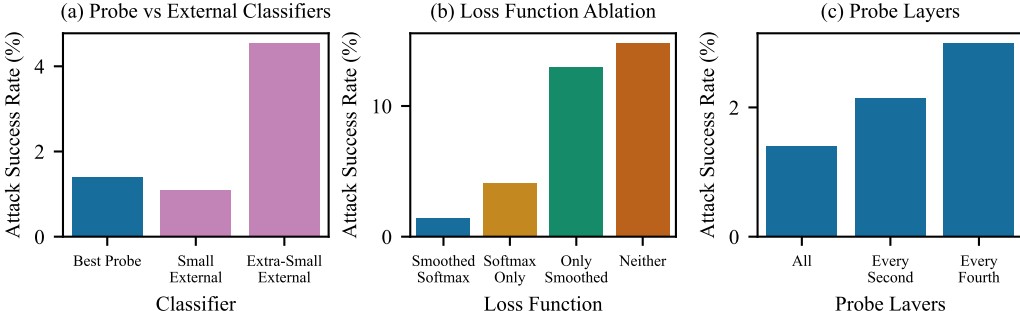

Figure 2: **Analysis of linear probe performance for CBRN-related harmfulness detection using static jailbreak data. (a) Attack success rates for different models.** We find that probes achieve competitive robustness to fine-tuned Constitutional Classifiers. **(b) Ablation study of loss function.** Combining logit smoothing and softmax pooling yields the best performance. **(c) Impact of layer selection on probe performance.** Performance degrades as fewer layers are utilized for probing. All evaluations were conducted on static jailbreak datasets of CBRN-related queries. All classifiers are exchange classifiers and are calibrated to 0.1% refusal rates on WildChat.

This method is closely related to the softmax probes of McKenzie et al. (2025), which also route gradients toward tokens indicative of the positive class via softmax pooling. However, we explicitly target streaming classification rather than making a single prediction for each sequence of tokens, and McKenzie et al. (2025) do not employ logit smoothing.

## 5.2 ANALYSIS

We now analyze the impact of various probe design choices on classifier performance, and compare the performance of linear probes with separate, fine-tuned classifiers.

**Training details.** We train a linear probe on top of a Claude Sonnet 4 using concatenated activations across all layers. For the exchange dataset, we employ the same training data as in the previous section but replace the synthetic pipeline's hard labels with exchange-level soft labels produced by a Constitutional Classifier fine-tuned from Claude Sonnet 4.[6] For probe training, we use softmax temperature $\tau = 1$, window size $M = 16$, and train on ca. 25% of the full exchange dataset; we found training on more examples did not improve probe performance. For evaluation, we compute predictions using an exponentially weighted moving average. As baselines, we compare against the first-stage and second-stage classifiers from Section 4. However, in this section, both of these classifiers are calibrated to a refusal rate of 0.1% on WildChat (Zhao et al., 2024).

**Jailbreak data.** We assess robustness using a dataset of jailbreaks collected from human red-teaming, comprising approximately 7,000 exchanges designed to elicit harmful CBRN-related information. We use only exchanges produced by red-teamers whose exchanges were never used for classifier training. Following Sharma et al. (2025), we evaluate response harmfulness using LLM-rubric grading supplemented with additional LLM filters to exclude refusals. We evaluate the harmfulness of the entire model completion using the rubric grader, but consider the attack successfully defended against if the exchange classifier flags at any point during model sampling.[7] For the analysis in this section, we truncate the datasets to have a maximum context length of 20K tokens.

**Probe robustness approaches external classifiers.** We first compare the attack success rate of our best-performing linear probe against the small (Claude Haiku 4.5) and extra-small external classifiers used in our two-stage system (Figure 2a). The probe achieves an attack success rate that outperforms the extra-small classifier and almost matches the Haiku 4.5 classifier. These results suggest that linear probes could potentially serve as viable first-stage classifiers in cascaded defense systems. Moreover, we found that the flag rate on benign scientific requests was higher for the probe.

---

[6]We found either distilling with soft-labels or using hard labels from our synthetic data pipeline leads to similar performance.

[7]This may underestimate the attack success rate.

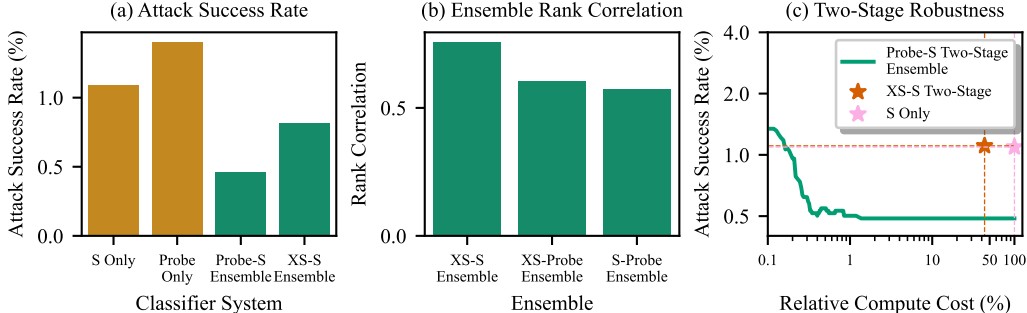

Figure 3: **Combining linear probes with external classifiers improves robustness and reduces costs. (a) Attack success rates for classifier systems.** We find that combining our probe and a small external classifier yields the best performance, outperforming the more expensive external-classifier ensemble. **(b) Spearman rank correlation between classifier predictions on jailbreak attempts.** Probes and external classifiers make more independent errors than pairs of external classifiers. **(c) Compute-robustness tradeoff curves for two-stage configurations.** We use the probe predictions to determine routing to the second-stage classifier, which uses the average logit across the probe and small classifier for prediction. We present the cost relative to using the small-sized classifier on all traffic. On our dataset of CBRN-related exchanges, our system that combines probes and the small external classifier can exceed the robustness of the small classifier alone whilst simultaneously offering a 100x reduction in compute costs. All systems are exchange classifiers calibrated to have a 0.1% refusal rate on WildChat. The linear probe uses activations from Claude Sonnet 4, which is the protected model. The 'S' classifier is fine-tuned from Claude Haiku 4.5, while the 'XS' classifier is fine-tuned from an internal model at approximately one-third the size. We evaluate robustness using LLM-based rubric grading on a dataset of CBRN-related human red-teaming exchanges.

**Softmax pooling and smoothing are critical for probe performance.** We assess the importance of softmax pooling and logit smoothing by training probes that do not use these (Figure 2b). Using neither smoothing nor softmax pooling results in the highest attack success rate. While applying either technique alone improves performance, combining both yields the best performance. We consider additional loss functions in Appendix Figure 5.

**Multi-layer probing substantially improves performance.** Our default probe uses all layers. To examine the importance of this choice, we consider the performance of probing on different numbers of layers (Figure 2c). We find that using all available layers achieves the lowest attack success rate at approximately 1.4%, while using fewer layers consistently yields worse performance.

## 5.3 COMBINING PROBES AND CLASSIFIERS

We now investigate whether combining probes and classifiers can offer further performance improvements. Indeed, ensemble methods have a long history in machine learning at improving performance and robustness (e.g., Dietterich, 2000; Lakshminarayanan et al., 2017).

**Ensembling improves robustness.** We explore weighted ensembles between our classifiers (Figure 3a). We evaluate robustness using the dataset, evaluation method, and calibration procedure described previously. For each classifier pair, we combine their logits using a weighted average: $z_{\text{ensemble}} = \alpha z_1 + (1 - \alpha)z_2$, where $\alpha$ is optimized per pair to minimize attack success rate on the test set. Ensembling the probe with the small external classifier achieves the lowest attack success rate, substantially outperforming either classifier alone. Notably, equal weighting ($\alpha = 0.5$) proves near-optimal for this ensemble—see Appendix Figure 6. While combining the extra-small and small external classifiers yields some improvement over individual classifiers, the gains are modest compared to the probe-based ensemble.

**Probes provide complementary predictions to external classifiers.** To understand why probe-classifier ensembles outperform classifier-only ensembles, we analyze the correlations between classifier predictions on our red-teaming exchange dataset (Figure 3b). We find the Spearman rank cor-

relation between the probe and the small external classifier is lower than that between the pair of external classifiers. This suggests that despite being trained on the same distribution as the external classifiers, the probe appears to leverage signals that complement those used by external classifiers, explaining the effectiveness of probe-classifier ensembling.

**Probe-classifier cascades reduce computational overhead.** Finally, we examine whether probes can improve the compute-robustness tradeoff in two-stage classifier cascades. We use our lightweight probe continuously during generation and invoke a more expensive second-stage classifier selectively based on the probe's confidence. Specifically, when the probe score exceeds a threshold, we route the exchange to the small external classifier and average their logits to make our final prediction. Otherwise, we rely solely on the probe. By varying the probe threshold, we control the fraction of exchanges requiring external classification and thus the compute cost. A per-layer probe is structurally a single extra MLP neuron, so if implemented efficiently the overhead is $\mathcal{O}(1/d_{\mathrm{MLP}})$—typically $\sim 10^{-4}$ or less, so cascade cost is almost entirely the external classifier.

We find the probe-classifier cascade substantially improves both efficiency and robustness relative to the two-stage system described in Section 4, and also compared to a single exchange classifier (Figure 3c). On our dataset, our system exceeds the robustness of the small classifier alone whilst simultaneously offering a 100x reduction in compute costs.

# 6 PRODUCTION-GRADE CONSTITUTIONAL CLASSIFIERS

This section presents our final production-grade Constitutional Classifier system, which integrates the techniques from preceding sections to achieve strong robustness, low false positive rates, and reduced computational overhead.

**System architecture.** The production-grade system uses a weighted two-stage classifier cascade. A lightweight linear probe monitors all exchanges during generation. When the probe score exceeds a threshold, the exchange is escalated to a second-stage external classifier. For escalated exchanges, the final classification score is computed as a weighted combination of probe and classifier logits: $z_{\mathrm{final}} = 0.55 \cdot z_{\mathrm{probe}} + 0.45 \cdot z_{\mathrm{classifier}}$. For the second-stage external classifier, we fine-tune a Constitutional Classifier from Claude Haiku 4.5. The system described here is used to guard Claude Sonnet 4.5. As such, the probe uses activations from that model.

**Deployment metrics.** We evaluated the system through a shadow deployment on production traffic using Claude Sonnet 4.5 from December 1, 2025, to January 1, 2026. The flag rate—the proportion of queries refused—was 0.05%, substantially lower than the 0.38% reported by Sharma et al. (2025). The first-stage probe escalated approximately 5.5% of traffic to the second-stage classifier. In terms of computational overhead, this represents roughly an 8x reduction compared to the two-stage system described in Section 4, and approximately a 40x reduction compared to the single exchange classifier in Section 3.

**Robustness results.** We applied the same red-teaming methodology described in previous sections. The system underwent an estimated 1736 cumulative hours[8] of red-teaming effort across approximately 198K attempts. Only one high-risk vulnerability was discovered, corresponding to a detection rate of 0.005 per thousand queries—the lowest among all systems considered in this paper. The discovery time for this attack was 30 hours, representing the highest minimum vulnerability discovery time observed. No universal jailbreak was found capable of answering all eight target questions at half the level of detail provided by a model without safeguards. These results represent the strongest robustness achieved across all systems evaluated in this work.

# 7 RELATED WORK

**Adaptive computational schemes.** Recent work has explored adaptive two-stage classification approaches for content moderation. OpenAI's systems (OpenAI, 2025a;b) employ a lightweight topic filter to identify high-harm content areas, triggering more expensive classification only when necessary. While their first stage focuses on topic detection, our approach explicitly targets jailbreak

---

[8]Estimated using API logs, following the methodology of Sharma et al. (2025) but with 5-minute buckets.

Table 1: **Comparison of Constitutional Classifier systems evaluated in this work**. We measure the compute overhead relative to the last-generation system. We measure robustness using the vulnerability discovery rate, the number of high-risk vulnerabilities discovered per thousand red-teaming queries. Production traffic refusal rate is the percentage of production queries refused by the system. Lower is better for all metrics. The production-grade system described in this section achieves the best trade-off between robustness, inference overhead, and refusal rates.

| System | Relative Compute Overhead (%) | High-Risk Vulnerability Discovery Rate | Production Traffic Refusal Rate (%) |
|---|---|---|---|
| Last Generation (§2) | 100.0 | 0.01871 | 0.073 |
| Exchange Classifier (§3) | 150.0 | 0.00885 | 0.038 |
| Two-Stage Cascade (§4) | 27.8 | 0.00878 | **0.036** |
| Production Grade (§6) | **3.5** | **0.00505** | 0.050 |

attempts in the initial stage.[9] Hua et al. (2025) investigate optimal strategies for combining multiple monitors under cost constraints. In contrast, we implement a straightforward two-stage classification scheme and validate its robustness through extensive human red-teaming. These approaches reflect broader trends in adaptive computation, including systems like TARS (Kim et al., 2025), which dynamically allocate test-time compute based on query complexity. Other work explores cascades and model routers for efficient LLM deployment. HybridLLM (Ding et al., 2024) employs a lightweight BERT-style decoder to route queries between small and large LLMs, AutoMix (Aggarwal et al., 2025) adaptively combines outputs across models using confidence scores and learned routing, and recent advances have refined cascades through confidence-based deferral (Rabanser et al., 2025).

**Model-internals approaches.** Several methods use model internals for efficient classification. Cunningham et al. (2025) explore internals-based classifiers, focusing on last-N layer networks, while we focus on boosting the performance of simple linear probes with softmax pooling and logit smoothing. McKenzie et al. (2025) similarly investigate softmax-pooled probes and classifier cascades. While they aggregate token scores into single sequence-level predictions, we build streaming classifiers with continuous predictions during generation. Our analysis in Section 5 shows that combining softmax pooling with logit smoothing substantially improves performance over either alone. Beyond standard activation probes for harmfulness detection (Alain & Bengio, 2016; Zou et al., 2023; Youstra et al., 2025), recent approaches fine-tune LLMs using internals-based losses, including short-circuiting (Zou et al., 2024) and latent adversarial training (Casper et al., 2024). Others build classifiers using sparse auto-encoder features (Bricken et al., 2024; Kantamneni et al., 2025).

# 8    CONCLUSION

Our work demonstrates that Constitutional Classifiers can achieve production-grade jailbreak robustness with dramatically improved deployment viability. Our approaches include exchange classifiers that evaluate outputs within their conversational context to prevent obfuscation attacks, cascaded classifiers that reserve expensive classification only for flagged content, and activation-based probes. We develop systems that provide robust protection against universal jailbreak attempts while meeting the stringent false positive and computational constraints required for deployment, thus establishing Constitutional Classifiers as practical and effective safeguards for production LLMs.

**Future work.** Tighter integration between classifier safeguards and language models—for example, incorporating classifier signals directly into model sampling processes, or training models to better resist obfuscation attempts—could strengthen robustness. Additionally, improving training data through automated red-teaming (Perez et al., 2022) or including data more representative of production traffic could yield better classifiers.

---

[9]As noted by OpenAI (2025b): "The first tier in this system is a fast, topical classifier model that determines whether or not the content is related to biology."

## ACKNOWLEDGEMENTS

We thank Xander Davies, Robert Kirk, Ben Edelman, and Holden Karnofsky for valuable feedback and discussions. We thank Andrew Wang and Jenny Bao for identifying the discrepancy between the implemented training objective and its description in an earlier draft and testing its impact respectively. The robustness analysis of our system through human red teaming was made possible by the dedicated efforts of our red-teamers and substantial operational support from HackerOne. We also thank UK AISI, US CAISI, and FAR.AI for red teaming various versions of our system. Mrinank Sharma thanks Rob Burbea for foundational inspiration, guidance, and support.

## AUTHOR CONTRIBUTIONS

**Probe Development.** Hoagy Cunningham, Andrew Persic, and Jordan Abderrachid were the core contributors to probe development. Hoagy Cunningham led probe methodology and evaluation; Andrew Persic led infrastructure development; Jordan Abderrachid made significant contributions to both areas.

**Exchange and Two-Stage Classifiers.** Jerry Wei led the work on exchange classifiers and two-stage classifiers, with substantial assistance from Zihan Wang and Alwin Peng.

**Red-Teaming.** Logan Howard and Giulio Zhou conducted classifier red-teaming. Clare O'Hara supported the red-teaming program.

**Infrastructure and Monitoring.** Austin Cohen supported classifier monitoring in real-world traffic. Jin Pan, Rob Gilson, Yue Song, Rohit Mittapalli, Alek Dimitriev, Bobby Chen, Christopher Liu, Yijin Hua, Andy Dau, Andrew Persic, Jordan Abderrachid, Xunjie Yu, Alex Silverstein, and Raj Agarwal contributed to probe and classifier infrastructure implementations.

**Leadership and Supervision.** Nikhil Saxena and Mu Lin provided management support. Vlad Mikulik provided additional management support. Jared Kaplan and Jan Leike provided high-level guidance. Mrinank Sharma, Ethan Perez, and Jerry Wei contributed to project supervision. Mrinank Sharma wrote the majority of the paper, with feedback from other authors.

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

# A  DETAILED SYSTEM COMPARISON

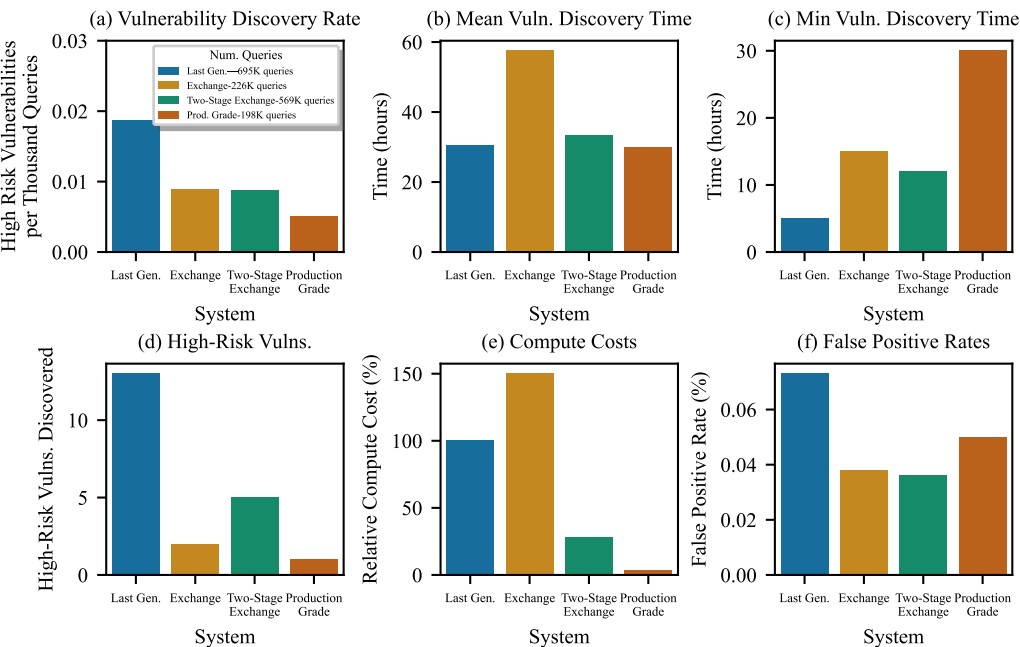

Figure 4: **Comprehensive comparison of Constitutional Classifier systems across robustness, computational efficiency, and false positive rates.** (a) High-risk vulnerability discovery rate normalized per thousand queries, with total query counts shown for each system. (b) Mean time in hours for discovering high-risk vulnerabilities. (c) Minimum time to first vulnerability discovery. (d) Absolute count of high-risk vulnerabilities discovered. (e) Relative computational cost compared to our implementation of the last-generation defense system. (f) Refusal rates on production traffic, measured using the refusal rates in the first week after deployment to production traffic. The production-grade system achieves the best robustness and computational efficiency with acceptable refusal rates.

In this section, we provide a comprehensive comparison of four Constitutional Classifier systems evaluated in this work through both deployment and human red-teaming: (i) our implementation of the last-generation dual-classifier system with separate input and output classifiers; (ii) the single exchange classifier system; (iii) a two-stage exchange classifier cascade with updated classifiers and inference optimizations; (iv) our production-grade classifier system, which uses a linear probe as a first-stage classifier, and a probe-external exchange ensemble as the second stage. Figure 4 presents key metrics for these systems.

We emphasize that we use the **high-risk vulnerability discovery rate** as our primary robustness metric to account for the varying number of queries submitted across different red-teaming campaigns. The other metrics we consider are expected to have a much stronger dependence on the volume of red-teaming, which varies across systems. We define a high-risk vulnerability as any attack that successfully answers more than five of eight target questions with rubric scores at least half that of a model without safeguards. The discovery rate is then calculated as the number of high-risk vulnerabilities identified per thousand red-teaming queries submitted, normalizing for the different levels of red-teaming effort applied to each system.

To calculate the total red-teaming queries, we calculate the number of red-teaming queries submitted by users whose queries have a classifier refusal rate of greater than 5%. This filters out queries made by users who might be using our red-teaming bounty program for purposes other than red-teaming.

## B  FURTHER DETAILS ON HUMAN RED-TEAMING

We evaluate the robustness of our Constitutional Classifier systems through extensive human red-teaming, largely following the protocol established by Sharma et al. (2025).

### B.1  OVERALL APPROACH

Our red-teaming protocol provides participants with target queries and instructs them to develop universal jailbreak strategies—prompting techniques that bypass safeguards and elicit detailed responses across all target queries. We use eight target queries spanning various harmful categories and assess response quality using LLM-based rubric grading.

**Evaluating jailbreak success.** Following Sharma et al. (2025), we implement a multi-stage LLM-based rubric-grading pipeline. We first generate baseline outputs from models without safeguards to establish the level of detail expected for each query. These baselines are used to create query-specific rubrics containing key information elements. Candidate jailbreak responses are evaluated against these rubrics, with a response considered successful if it achieves at least 50% of the score obtained by a model without safeguards. We define a *high-risk vulnerability* as any attack that successfully answers more than five of eight target questions at this threshold.

**Red-teaming structure.** We provide red-teamers with API access to the defended model. Participants submit their jailbreak attempts along with self-reported development time. Throughout this work, we offered bounties scaled by the number of successfully jailbroken queries, with maximum payouts ranging from $25K to $35K USD depending on the specific campaign. To calculate the total red-teaming queries, we calculate the number of red-teaming queries submitted by users whose queries have a classifier refusal rate of greater than 5%. This filters out queries made by users who might be using our red-teaming bounty program for purposes other than red-teaming.

### B.2  PROTOCOL MODIFICATIONS FOR TWO-STAGE SYSTEM

For the two-stage exchange classifier evaluation and production-grade system, we implemented several protocol refinements to improve red-teaming efficiency.

**Program structure changes.** We transitioned from providing API credits to granting access through a specific model alias that does not consume credits, reducing friction for participants and preventing misuse of our platform. We further provided participants with test queries and rubrics, which we found improved submission quality. The universal jailbreak definition was clarified to emphasize the need for detailed, actionable responses across multiple queries.

**Reward structure updates.** We restructured the bounty system to better incentivize universal jailbreaks. The maximum reward was increased to $35K USD. We further included a 1.05x multiplier for each additional model response considered to be a "strong pass", which is a response achieving at least 75% of the detail of a model without safeguards, as assessed by rubric grading. This incentivizes red-teamers to improve the details provided by their responses, as well as the number of questions answered. We also increased rewards for universal jailbreaks and reduced rewards for attacks that were less universal (that answered fewer queries), to further incentivize the discovery of universal attacks.

**Question set refinement.** We replaced four of the original eight target queries with newer questions. We provided the previous questions and rubrics to red-teamers to allow them to improve their submissions.

**Data collection improvements.** We implemented standardized data collection procedures and added explicit terms regarding data storage and usage. Participants were required to document their attack development process and provide time estimates.

## C  ADDITIONAL PROBE ABLATIONS

We now conduct further ablation studies to better understand the impact of different loss function designs and hyperparameters on probe performance.

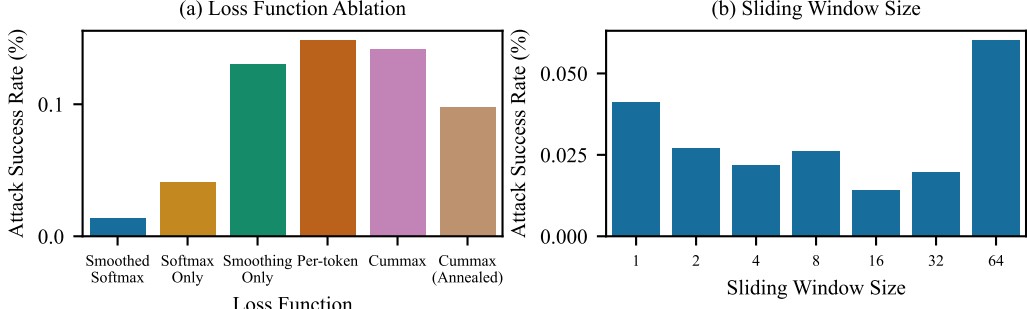

Figure 5: **Additional probe ablations on loss functions and hyperparameters.** (a) Comparison of different loss functions showing that our smoothed softmax approach additionally outperforms cumulative maximum and annealed cumulative max loss, which were proposed by Sharma et al. (2025). (b) Impact of sliding window size $M$ on attack success rate, demonstrating optimal performance at moderate window sizes (around $M = 16$), with degradation at both extremes.

## C.1 ALTERNATIVE LOSS FUNCTIONS

Beyond the smoothed softmax loss presented in the main text, we investigate two additional loss formulations that leverage the streaming nature of our classification task.

**Cumulative maximum loss.** We explore using the cumulative maximum of probe probabilities as the predictor for the full sequence label. This approach better reflects the streaming use case where, if we stop generation at position $t$, the remainder of the sequence is effectively considered harmful. Specifically, we define:

$$p(y = 1|x_{1:t}) = \max_{\tau \leq t} \sigma(\bar{z}_\tau(x_{1:\tau})) \tag{5}$$

where $\bar{z}_\tau$ is the probe logit at position $\tau$ (after sliding window averaging). This ensures that predictive probabilities only increase when the probe is confident that the full sequence will be harmful. However, using this approach, gradients only flow through the maximum-scoring token position, which can mean this approach sometimes suffers from optimization challenges.

**Annealed cumulative maximum loss.** To address the optimization challenges of the cumulative maximum while retaining its benefits for streaming classification, we linearly interpolate between direct probe probabilities and the cumulative maximum throughout training, as suggested by Sharma et al. (2025):

$$p(y = 1|x_{1:t}) = (1 - \omega) \cdot \sigma(\bar{z}_t(x_{1:t})) + \omega \cdot \max_{\tau \leq t} \sigma(\bar{z}_\tau(x_{1:\tau})) \tag{6}$$

where $\omega$ starts at 0 and linearly increases to 1 throughout training. This ensures stable initial training with good gradient flow while gradually transitioning to the cumulative maximum formulation that better matches our streaming use case.

Figure 5a compares these loss functions against our proposed smoothed softmax approach. We find that our softmax pooling with logit smoothing achieves stronger performance than these additional loss functions. We use the same evaluation datasets and methodology as described in the main paper.

## C.2 IMPACT OF SLIDING WINDOW SIZE

The sliding window size $M$ controls the degree of smoothing applied to probe logits. Figure 5b demonstrates that moderate smoothing is crucial for performance. Performance improves as we increase the window size from no smoothing, reaching optimal performance around $M = 16$. However, excessive smoothing degrades performance.

## D    ENSEMBLE WEIGHTING ANALYSIS

Figure 6 examines three classifier pairs and shows how their individual performance varies with different weighting proportions. We use a weighted average of logits for each classifier pair. While we optimize $\alpha$ for each classifier pair in Figure 3a, for the classifier-probe ensembles, we find that equal weighting already is close to optimal.

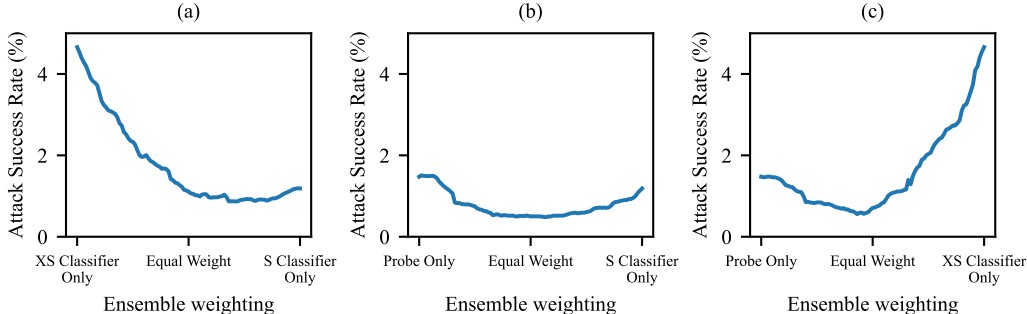

Figure 6: **Impact of ensemble weighting on attack success rates for different classifier pairs.** Each panel shows the performance of individual classifiers and logit-weighted ensembles. (a) Extra-small (XS) and small (S) external classifiers. (b) Linear probe and small external classifier. (c) Linear probe and extra-small external classifier. All classifiers are calibrated to maintain a 0.1% false positive rate on WildChat. We use our standard dataset of red-teaming exchanges and our standard evaluation methods, as described in the main paper.

## E    LESSONS LEARNED FROM CLASSIFIER ITERATION AND DEPLOYMENT

We now share lessons learned from classifier iteration and deployment.

**Infrastructure reliability is a security requirement.** During development and human red-teaming, we discovered that infrastructure bugs can create effective jailbreaks. Comprehensive end-to-end testing is therefore essential to ensure consistency across different implementations and deployment environments. Even perfect classifiers become ineffective when compromised by implementation errors or configuration issues.

**Regular red-teaming remains essential.** Our robustness evaluations demonstrate that periodic human red-teaming must complement synthetic testing. While automated evaluations enable rapid iteration and improvement, relying solely on synthetic benchmarks can lead to overfitting on limited test cases. Organizations must therefore invest in both technical infrastructure and operational processes that enable efficient deployment, testing, and evaluation of updated safety systems.

**On-the-fly activation computation accelerates probe development.** During probe classifier experimentation, we found that recomputing model activations *within* the training loop rather than pre-computing and saving them offers compelling engineering advantages. Moving probe activation data from high-bandwidth memory (HBM) to RAM or blob storage creates severe I/O bottlenecks that dwarf the computational cost of regeneration. Furthermore, since linear probe training is extremely efficient, we can test multiple probe variants simultaneously on freshly computed activations.

**Providing red-teamers with test queries is helpful.** Supplying red-teamers with sample questions and their corresponding evaluation rubrics significantly enhances red-teaming efficiency. These examples guide red-teamers in developing appropriately concerning jailbreaks, thereby streamlining the report evaluation process.

## F    DETAILS OF LLM CONTRIBUTION

We used LLMs to aid and polish paper writing.

