# OpenReview forum: "Constitutional Classifiers++: Efficient Production-Grade Defenses against Universal Jailbreaks"
_ICLR.cc/2026/Conference — ICLR 2026 Poster_

### Official Review · Reviewer_HfNu · 2025-10-24

**Soundness:** 3
**Presentation:** 3
**Contribution:** 3
**Rating:** 6
**Confidence:** 3

**Summary:**

This paper introduces "Constitutional Classifiers++," a jailbreak defense system that is both significantly more robust and 5.4x more computationally efficient than prior work. The authors first identify that classifiers evaluating inputs and outputs in isolation are vulnerable to "reconstruction" and "obfuscation" attacks. They address this by introducing an "exchange classifier" that evaluates the model's output in the context of the full conversational input. To manage this method's high cost, they implement a two-stage "classifier cascade" where a cheap, lightweight classifier screens all traffic and escalates only suspicious queries to the stronger, more expensive one . This system is validated against over 560K red-teaming queries, demonstrating high robustness and a 10x lower refusal rate on production traffic. The paper also proposes using linear activation probes, trained with a novel softmax-weighted loss and logit smoothing, as a highly efficient future direction for the first-stage classifier .

**Strengths:**

1. The validation of the primary defense system against over 560K human red-teaming queries is a massive, high-quality evaluation that provides extremely strong evidence of robustness against adaptive attackers.
2. The paper is written with a clear, compelling narrative. It begins by demonstrating a concrete failure in existing systems (Section 2, Figure 1), then systematically presents the solutions to fix the vulnerability (Section 3, exchange classifier) and its associated cost (Section 4, cascade). This structure makes the paper's contributions easy to understand.
3. The paper reports not just on robustness, but on a 5.4x computational cost reduction and a 10x reduction in false positives (0.036% refusal rate), all backed by production deployment data . This directly addresses the most critical barriers to deploying LLM defenses at scale.

**Weaknesses:**

1. The paper's core thesis, established powerfully in the introduction and Sections 3 and 4, is that defenses must be validated against large-scale, adaptive human red-teaming to make credible robustness claims. However, the highly promising results for the linear probes—which are presented as a key part of the contribution—are validated only on a static, 7,000-example dataset.
2. This makes the claims about the probe-classifier cascade's "100x reduction in compute costs" (Figure 3c)  feel premature. We have no evidence that this probe-based system would survive the 560K-query red-teaming that the main system was subjected to.

**Questions:**

Question-1.: Figure 3a shows the Probe-S ensemble is the most robust system on your static dataset, and Figure 3b suggests this is because the probe and the external classifier have low correlation and catch "complementary" attacks. Could you provide any qualitative examples of what kinds of attacks the probe catches that the external classifier misses, and vice versa? Understanding this complementary nature seems critical to assessing whether the probe is truly adding a new, generalizable layer of defense or just overfitting to this specific dataset.

---

> ### Author Response · Authors · 2025-11-17
>
> We thank the Reviewer HfNu for their insightful review and particularly for identifying the probe validation gap. We've made a number of clarifications below, which we hope will allow you to increase your evaluation more. Moreover, we've conducted new red-teaming on the probe, addressing one of the major weaknesses.
>
> > The paper's core thesis, established powerfully in the introduction and Sections 3 and 4, is that defenses must be validated against large-scale, adaptive human red-teaming to make credible robustness claims. However, the highly promising results for the linear probes—which are presented as a key part of the contribution—are validated only on a static, 7,000-example dataset.
>
> Thank you for sharing this well-founded and highlights an important methodological point. **We now have an extensive adversarial validation for the probe system**. We used a classifier cascade, using a linear probe for the first layer, and an external classifier for the second layer. There were 157K attack attempts and 194 red-teamers. We found one universal jailbreak, however, this jailbreak was actually flagged by the probe and then let through by the external classifier, supporting that probes can act as first-stage classifiers while achieving the computational cost savings.
>
> > This makes the claims about the probe-classifier cascade's "100x reduction in compute costs" (Figure 3c) feel premature. We have no evidence that this probe-based system would survive the 560K-query red-teaming that the main system was subjected to.
>
> Thank you for sharing this. We believe with the additional round of red-teaming conducted above, we are confident in the ability of the probe-based system to provide sufficient robustness. However, we also acknowledge that the original claim may have been premature, and are happy to improve the writing in the camera-ready version of the paper to tighten this up.
>
> > Could you provide any qualitative examples of what kinds of attacks the probe catches that the external classifier misses, and vice versa? Understanding this complementary nature seems critical to assessing whether the probe is truly adding a new, generalizable layer of defense or just overfitting to this specific dataset.
>
> Thank you for sharing this. In general, the probe and external classifier operate on different information—internal activations versus semantic text content—leading to complementary detection and different inductive biases. We haven't been able to find clear, systematic patterns in generalization differences, but we agree this would be an interesting avenue for future work. With regards to other attack patterns in external classifiers, we've found:
> * One attack pattern is "flooding output," where attackers elicit responses containing large amounts of general information rather than specific harmful content. Because of the way the rubric grading works, these responses can sometimes include rubric keywords.
> * Prompt injection attempts insert hidden instructions within prompts to manipulate classifier behavior. While such attacks occasionally bypass exchange classifiers, they can be mitigated through improved training data curation.
> * Other attacks combine multiple techniques, like ciphers, roleplaying, other languages, sometimes with very long context.
> We'll look into this more for the camera ready version, and will discuss this more in the revised paper.

---

> ### Comment · Reviewer_HfNu · 2025-11-24
> **Reply to Authors**
>
> Thanks for the Prompt rebuttal. I have increased my score now.
> Best of Luck!!

---

### Official Review · Reviewer_e8Wf · 2025-10-28

**Soundness:** 3
**Presentation:** 2
**Contribution:** 3
**Rating:** 4
**Confidence:** 3

**Summary:**

This paper introduces a series of new constitutional classifiers that achieve a high level of jailbreak robustness while reducing computational costs. The proposed classifiers build upon the last generation of constitutional classifiers, whose goal is to defend against most universal jailbreak attacks. The authors identify that existing classifiers are vulnerable to two types of attacks: reconstruction and output obfuscation attacks.

**Strengths:**

- Important and timely topic
- Strong contribution in enhancing classifier performance and reducing computational cost
- Practical insights for industry applications

**Weaknesses:**

- Lack of transparency and clarity
- Unclear release plan

**Questions:**

Thank the authors for submitting this work. I believe this paper makes a strong contribution toward developing a robust and efficient classifier to defend against universal jailbreak attacks on LLMs. In detail, the main contributions can be summarized as follows:

(1) Identification of two critical vulnerabilities in existing classifiers and the introduction of exchange classifiers to improve robustness;

(2) A two-stage classification framework to reduce the computational cost of exchange classifiers;

(3) The introduction of linear probes that reuse model representations to further reduce inference cost.

Despite these strengths, I find it difficult to fully understand the technical details without carefully reading the original *Constitutional Classifiers* paper, as much of this work builds directly upon it (e.g., dataset construction, evaluation pipeline, and red-teaming protocol).

Based on this, my main concerns are as follows.

### **1. Lack of transparency and clarity**

Many key aspects are described only at a high level, making it challenging to evaluate technical correctness.

- For instance, after reading Section 3, I still find the definition and mechanism of the exchange classifier unclear. Does it operate alone, or does it replace the input classifier while being used together with the output classifier? Its relevant descriptions, such as “exchange classifier uses a small-size LLM to evaluate outputs, rather than an extra-extra-small LLM” and “internal LLMs,” are also ambiguous and lack of necessary clarity.
- Another example is the discrepancy between the 569 K and 226 K query counts in the robustness results of Section 3. It is not explained how these datasets are constructed or sampled and why different classifiers are evaluated on different testbeds.
- Such general and ambiguous descriptions are quite common throughout the paper.

### **2. Limited reproducibility**

Most of the experiments rely on internal proprietary models, datasets, and costly human red-teaming. The paper does not indicate whether any of these materials, or even synthetic evaluation data, will be released. Given ICLR’s emphasis on open and reproducible research, this limitation could weaken the work’s transparency and community impact.

---

> ### Author Response · Authors · 2025-11-17
>
> We thank Reviewer e8Wf for recognizing the importance of our work and providing detailed concerns about transparency. We are pleased that the reviewer judged this work to be important, timely, and making a strong contributions. We've made a number of clarifications below, which we hope will allow you to raise your evaluation score, and are commited to improving the camera ready version of the paper to improve the writing clarity.
>
> # 1. Lack of transparency and clarity
>
> > Many key aspects are described only at a high level, making it challenging to evaluate technical correctness. For instance, after reading Section 3, I still find the definition and mechanism of the exchange classifier unclear. Does it operate alone, or does it replace the input classifier while being used together with the output classifier? Its relevant descriptions, such as “exchange classifier uses a small-size LLM to evaluate outputs, rather than an extra-extra-small LLM” and “internal LLMs,” are also ambiguous and lack of necessary clarity.
>
> Thank you for raising this, and we apologize around the lack of clarify. **We're happy to make improvements to the camera-ready version of the paper to address this**. We believe that this confusion stems from the difference between an "exchange classifier in general", and the exchange classifier system that we tested. Exchange classifiers in general can be used with or without output classifiers, but the system we tested specifically did not also use an output classifier. We admit this was confusing.
>
> For "internal LLMs", we mean large language models that we have pre-trained ourselves, and that are proprietary and not available. We cannot reveal more information about the sizes of these models.
>
> > Another example is the discrepancy between the 569 K and 226 K query counts in the robustness results of Section 3. It is not explained how these datasets are constructed or sampled and why different classifiers are evaluated on different testbeds.
> Such general and ambiguous descriptions are quite common throughout the paper.
>
> We apologize for this confusion. The difference between 569K and 226K query counts is because we tested the systems in seperate red-teaming campaigns during system iteration and development. The methodology used for testing was consistent between different models. However, we cannot control precisely how many red-teamers sign up. The methodology is consistent: red-teamers were instructed to break the system using any means necessary.
>
> ## 2. Limited reproducibility
>
> > Most of the experiments rely on internal proprietary models, datasets, and costly human red-teaming. The paper does not indicate whether any of these materials, or even synthetic evaluation data, will be released. Given ICLR’s emphasis on open and reproducible research, this limitation could weaken the work’s transparency and community impact.
>
> Thank you for raising this concern. Due to proprietary concerns and the risk of aiding attackers and undermining defences that are used for national security risks, we cannot publicly release code and models publicly. However, we are working on sharing datasets and evaluation methodologies with the Frontier Model Forum to enable responsible replication. We cannot release model weights due to proprietary concerns and the risk of aiding attackers. We believe this balance—protecting weaponizable information while sharing defensive techniques—is appropriate for security research.
>
> Moreover, we note that the purpose of sharing this paper at ICLR and with the community is to help and share insights from system development and techniques for making these defences better. We believe the paper makes a valuable contribution to the community.

---

> > ### Comment · Reviewer_e8Wf · 2025-11-26
> >
> > Thanks for the rebuttal.
> > However, as the presentation requires improvement, I will keep my score.

---

> > > ### Author Response · Authors · 2025-11-26
> > >
> > > Thank you for the response. Would you be so kind as to specify what presentation improvements you would need to see to increase your score?

---

### Official Review · Reviewer_dTVX · 2025-10-29

**Soundness:** 2
**Presentation:** 2
**Contribution:** 2
**Rating:** 4
**Confidence:** 3

**Summary:**

This study aims to demonstrate the proposed Constitutional Classifiers as practical safeguards for large language models.

**Strengths:**

- Good motivation: A key strength of this work is that it accurately identifies the practical deployment bottlenecks of [1], specifically its significant computational cost and tendency toward over-refusal. The authors then systematically tackle these issues with a series of well-motivated and targeted methods.
- Strong performance: high defense success rate with little refusal rate; 5.4x computation overhead reduction.

[1] Mrinank Sharma, Meg Tong, Jesse Mu, Jerry Wei, Jorrit Kruthoff, Scott Goodfriend, Euan Ong, Alwin Peng, Raj Agarwal, Cem Anil, et al. Constitutional classifiers: Defending against universal jailbreaks across thousands of hours of red teaming. arXiv preprint arXiv:2501.18837, 2025.

**Weaknesses:**

**Presentation:**
- No illustrative figure about the proposed methods. The absence of illustrative figures or diagrams detailing the proposed system architecture and classifier cascade makes it difficult to fully grasp the methodological workflow and component interactions.

**Novelty:**
- While the manuscript effectively builds upon [1], it overlooks meaningful discussion and comparison with established input-output-filtering based defense methods. This omission, along with the absence of comparative evaluation, undermines the claimed novelty, making the proposed exchange classifier appear more as a rebranding of existing filtering paradigms than a substantive methodological advance.
- The proposed logit smoothing, while empirically beneficial, appears more as an engineering trick than a fundamental algorithmic contribution.

**Several descriptions without theoretical justification, experimental verification, or detailed explanation:**
- The quantitative results referenced in lines 173, 203, 230, 235, and 239 lack sufficient experimental context. The authors should provide comprehensive implementation details, including computational platform, test datasets, model specifications, and baseline configurations, to ensure reproducibility and facilitate meaningful evaluation of the reported performance.
- The reliance on proprietary internal models and datasets, as indicated in lines 220 and 434, limits the verifiability and generalizability of the claimed results. To strengthen the validity and reproducibility of this work, the authors should supplement these findings with comprehensive experiments using publicly available models and standardized benchmarks.
- Several descriptions remain ambiguous and require clarification. For instance, Line 177 should quantify the classifier's effectiveness with specific performance metrics, while Line 344 needs explicit documentation of data sources and formats. Additionally, the conclusion in Line 348 would be strengthened by presenting detailed ablation results demonstrating how performance scales with varying training data sizes.

**Minor concerns:**
- Typo in Line 43.

**Questions:**

- Does the replacement of the sliding window with an EMA during inference (Line 283) introduce a training-inference inconsistency? Have the authors conducted experiments incorporating EMA during training, which might yield improved results?
- As shown in Figure 2(c), defense performance continues to improve with finer-grained probing. Have the authors experimented with even more granular approaches, such as inner-layer probing at the level of individual attention or MLP blocks?

---

> ### Author Response · Authors · 2025-11-17
>
> We thank Reviewer dTVX for the detailed feedback. We've made a number of clarifications below, which we hope will allow you to raise your evaluation score.
>
> ## Response to Weaknesses
> > No illustrative figure about the proposed methods. The absence of illustrative figures or diagrams detailing the proposed system architecture and classifier cascade makes it difficult to fully grasp the methodological workflow and component interactions.
>
> Thank you, we agree. We will add a system architecture diagram showing the exchange classifier, two-stage cascade, and linear probe positioning for the camera-ready version.
>
> > While the manuscript effectively builds upon [1], it overlooks meaningful discussion and comparison with established input-output-filtering based defense methods. This omission, along with the absence of comparative evaluation, undermines the claimed novelty, making the proposed exchange classifier appear more as a rebranding of existing filtering paradigms than a substantive methodological advance.
>
> Thanks for raising this. To clarify, the results in Section 3 are a comparative evaluation of the robustness of the established input-and-output classifier system, which we term the "last-generation system". Our extensive human red-teaming results show the exchange-classifier system is approximately 2x as robust. Given the large difference in robustness, we respectfully disagree that exchange classifiers are rebranding. Indeed, traditional input-output filtering evaluates components in isolation, making it vulnerable to reconstruction attacks (harmful instructions fragmented across benign segments) and output obfuscation attacks (harmful content in metaphorical language that appears benign without context). Our exchange classifier evaluates the output in the context of the full conversational input, enabling detection of these attacks. This joint evaluation is a fundamentally different approach.
>
> > The proposed logit smoothing, while empirically beneficial, appears more as an engineering trick than a fundamental algorithmic contribution.
>
> While it may appear as an engineering detail, logit smoothing addresses a fundamental challenge: individual token predictions are noisy and unreliable. Smoothing across windows provides stable, reliable classification during streaming inference. Combined with softmax-weighted loss, we believe this is a principled solutions crucial for performance, which we **empirically show is crucial for performance**. Therefore, we believe that this loss function is an important and valuable contribution.
>
> > The quantitative results referenced in lines 173, 203, 230, 235, and 239 lack sufficient experimental context. The authors should provide comprehensive implementation details, including computational platform, test datasets, model specifications, and baseline configurations, to ensure reproducibility and facilitate meaningful evaluation of the reported performance.
>
> We will add comprehensive details including computational platform,  model architectures, and baseline configurations for all reported results during the camera-ready stage, but are unable to do so at the moment, because we must preserve anonymity.
>
> > The reliance on proprietary internal models and datasets, as indicated in lines 220 and 434, limits the verifiability and generalizability of the claimed results. To strengthen the validity and reproducibility of this work, the authors should supplement these findings with comprehensive experiments using publicly available models and standardized benchmarks.
>
> Due to proprietary concerns and the risk of aiding attackers, we cannot publicly release code and models. We are working on sharing datasets and evaluation methodologies with the Frontier Model Forum to enable responsible replication.
>
> > Several descriptions remain ambiguous and require clarification. For instance, Line 177 should quantify the classifier's effectiveness with specific performance metrics, while Line 344 needs explicit documentation of data sources and formats. Additionally, the conclusion in Line 348 would be strengthened by presenting detailed ablation results demonstrating how performance scales with varying training data sizes.
>
> ## Response to Questions
> Q1 - EMA Training: We have not tested EMA during training. We use sliced means during training (more straightforward) and EMA during inference.
>
> Q2 - Finer-grained Probing: We have not experimented with attention-block or MLP-block level probing. Given that layer-level probing already achieves our targets, we prioritized deployment. This represents future work, which we'd be happy to discuss in the camera ready versoin.

---

> > ### Author Response · Authors · 2025-11-21
> >
> > Dear Reviewer,
> >
> > We also wanted to let you know that we experimented with using a simple MLP probe with width 512 using our dataset of red-teaming prompts. We found that the MLP probe did NOT improve performance over the linear probe. On a different version of the dataset, both methods achieve 99.9% TPR at 0.05%FPR on WildChat. Therefore, given the probe is cheaper, we prefer it.
> >
> > Thank you.

---

> > ### Comment · Reviewer_dTVX · 2025-11-26
> >
> > > Our extensive human red-teaming results show the exchange-classifier system is approximately 2x as robust.
> >
> > Could you please clarify what exactly “robustness” refers to here and provide a bit more detail on how the 2× improvement was measured?

---

> ### Author Response · Authors · 2025-11-26
>
> Yes. Robustness is the effort required to find a working attack. Quoting the paper:
>
> > The last-generation Constitutional Classifier system exhibited 13 high-risk vulnerabilities across 695K red-teaming queries, yielding a vulnerability rate of 0.01871 per thousand queries. By comparison, the exchange classifier system demonstrated superior robustness with only 2 vulnerabilities across 226K queries, corresponding to a rate of 0.00885 per thousand queries.
>
> We define a "high risk" vulnerability as an attack that answers 6 or more  target questions with rubric scores meeting a rubric score threshold. Therefore, The 2x improvement is in terms of rates of attacks found per thousand queries, with lower rates corresponding to higher robustness.

---

### Official Review · Reviewer_VZzS · 2025-10-30

**Soundness:** 3
**Presentation:** 3
**Contribution:** 3
**Rating:** 6
**Confidence:** 3

**Summary:**

The paper discusses a crucial problem: identifying jailbreak prompts designed to evade LLM safety mechanisms. A conversational classifier (known as exchange classifiers) is introduced, particularly to address reconstruction and obfuscation attacks. This is orchestrated using two stages to reduce computational costs while using internal model activations (through logit smoothing and weighted loss) to improve detection. The evaluation effort involves a red team, and also shows lower computational costs compared to previous approaches.

**Strengths:**

- The problem considered is very novel, and underexplored in LLM safety. The paper addresses deployment viability to achieve production-grade defenses.
- The use of exchange classifier is clever to mitigate reconstruction and obfuscation jailbreaks. The scalable modular design improves computational overhead significantly, compared to existing frameworks.
- The evaluation is excellent with LLM-based rubric grading for quantitative assessment

**Weaknesses:**

- Though the methodology is well described, it would be useful to have more details on architectural and training details for reproducibility and generalizability.
- It would be beneficial to compare alternative approaches to the probe methodology such as sparse autoencoder signals etc.)
- The dataset used for evaluation primarily focus on CBRN-related jailbreaks and internal red-team benchmarks. However, it is unclear how these results translate to broader diverse threat domains such as misinformation, hate etc.
- It would be also useful to describe analysis aiding to understand why certain attacks fail under the exchange classifier approach. Interpretive results would be useful to the community at large.

**Questions:**

1. It is unclear how large is the context for each instances in the exchange classifier? What is the length of conversation? How does this affect performance?
2. How well does the softmax-weighted loss differ from using moving average or median, particularly in the case where spurious tokens are present which skews the distribution?
3. In section 5.1, the linear probe architecture mentions activation features. Could these activation functions be non-linear? If so, would the results still hold true?
4. It is unclear what is the meaning of “calibrating classifier threshold to correspond to a 0.1% refusal rate on WildCat”? How can this statement be interpreted with respect to overall performance?

**Details Of Ethics Concerns:**

The paper uses a red-teaming protocol

---

> ### Author Response · Authors · 2025-11-17
>
> We thank Reviewer VZzS for recognizing the novelty of our problem and the strength of our evaluation. We make a number of clarifications below that we hope will allow you to increase your evaluation score.
>
> ## Response to Weaknesses
> > Though the methodology is well described, it would be useful to have more details on architectural and training details for reproducibility and generalizability.
>
> Unfortunately, we are unable to share this information due to it being propertiary information.
>
> > The dataset used for evaluation primarily focus on CBRN-related jailbreaks and internal red-team benchmarks. However, it is unclear how these results translate to broader diverse threat domains such as misinformation, hate etc.
>
> Our CBRN focus aligns with Anthropic and OpenAI's prioritization for frontier model safety. CBRN represents extremely high-stakes harms where defensive failures carry catastrophic consequences. We believe demonstrating production-grade defenses for the highest-risk category is the appropriate first contribution, while broader domain evaluation remains important future work.
>
> > It would be also useful to describe analysis aiding to understand why certain attacks fail under the exchange classifier approach. Interpretive results would be useful to the community at large.
>
> Interpretive Analysis: Thanks for sharing this information. We'd be happy to add some analysis of attack patterns that get past the exchange classifier:
> * One attack pattern is "flooding output," where attackers elicit responses containing large amounts of general information rather than specific harmful content. Because of the way the rubric grading works, these responses can sometimes include rubric keywords.
> * Prompt injection attempts insert hidden instructions within prompts to manipulate classifier behavior. While such attacks occasionally bypass exchange classifiers, they can be mitigated through improved training data curation.
> * Other attacks combine multiple techniques, like ciphers, roleplaying, other languages, sometimes with very long context.
>
> > It would be beneficial to compare alternative approaches to the probe methodology such as sparse autoencoder signals etc.)
>
> Thanks for sharing this. We focused on comparing against the classifier baselines, because these are the most commonly used safeguards and have been shown to perform well.
>
> ## Response to Questions
> **Q1 - Context Length**: The exchange classifier evaluates the full conversation histories, which vary substantially since attackers employ diverse strategies. We did not impose artificial constraints.
>
> **Q2 - Softmax-weighted Loss vs. Moving Average/Median:** We chose softmax weighting because it provides learned, adaptive weighting that emphasizes tokens most relevant for classification, while maintaining differentiability. **Figure 2 in the paper shows the smoothed-softmax weighing outperforms a moving average substantially**. The attack success rate for ``only-smoothed'' (which is a moving mean) is much higher than our loss function (~12% vs ~1.5%)
>
> **Q3 - Non-linear Probes:** We focused on linear probes due to their strong performance, interpretability, and efficiency. Since they already achieve our robustness targets, and are conceptually simple and easy to implement, we did not prioritize exploring non-linear alternatives. However, it is possible that non-linear probes can further improve performance. We'd be happy to include a discussion of this in the camera ready.
>
> **Q4 - Calibration Statement**: We set classifier thresholds such that only 0.1% of benign WildChat traffic is incorrectly blocked (false positives). This establishes our precision-recall trade-off. This is choosing the classifier score at which we block an output.

---

> > ### Comment · Reviewer_VZzS · 2025-11-27
> >
> > Thank you for the rebuttal, and answering most of my concerns. I have increased the score accordingly.

---

### Official Review · Reviewer_zAes · 2025-11-01

**Soundness:** 3
**Presentation:** 3
**Contribution:** 3
**Rating:** 6
**Confidence:** 3

**Summary:**

The paper introduces Constitutional Classifiers++, replacing the traditional input–output separation with exchange-level classification that understands conversational context. The system employs a two-stage cascade, a lightweight Stage-1 screener and a high-precision Stage-2 classifier to reduce runtime cost. It further integrates internal activation probes with external classifiers for ensemble robustness. Experiments report 5.4× cost reduction, a 0.036% block rate on production traffic, and full mitigation of universal jailbreak prompts across large red-team evaluations. Technical contributions include streaming linear probes with smoothed logits and softmax-weighted loss functions tailored for online inference.

**Strengths:**

1)	Clear motivation: The paper provides a clear motivation by diagnosing vulnerabilities in the prior Constitutional Classifier (CC). The authors identify two concrete and realistic attack classes: reconstruction attacks, in which harmful instructions are fragmented across benign segments, and output obfuscation attacks, in which malicious outputs are hidden behind metaphorical or coded language.

2)	Novelty: The paper presents two technically distinct yet complementary innovations that meaningfully advance safety mechanisms. First, the two-stage classification architecture introduces an adaptive cascade that balances robustness and efficiency. A lightweight first-stage model screens all interactions, while a stronger second-stage classifier verifies only flagged exchanges. This design reduces computational overhead by 5.4× without compromising jailbreak resistance, offering a deployable and scalable defense solution. Second, the linear activation probes represent a use of the LLM’s own internal activations for real-time harmfulness detection. By reusing in-model representations and applying techniques such as sliding-window logit smoothing and softmax-weighted loss, the probes achieve performance comparable to full Constitutional Classifiers at negligible cost.

3)	Training improvements: The study convincingly establishes that the proposed systems deliver cost-effective robustness. Both the two-stage cascade and the probe-based approach significantly reduce computational and training overheads while preserving or improving ASR performance, making them practical candidates for real-world, scalable LLM safety deployment.

**Weaknesses:**

1)	Analysis of the Failure Cases
While the exchange-classifier architecture clearly improves robustness relative to previous Constitutional Classifiers, the results in Section 3 suggest that failure cases remain and deserve deeper analysis. Specifically, the system still exhibited two high-risk vulnerabilities across 226K red-teaming queries (≈ 0.00885 per thousand), implying that some jailbreaks can still bypass contextual evaluation.

2)	Scaling trends for two-stage classifiers
While the paper provides strong empirical validation of the two-stage cascade architecture, it lacks a robustness trend on model scale, which limits the interpretability of the reported robustness gains. In Section 4, the two-stage setup combines “extra-small” and “small” classifiers, but the impact of scaling each stage independently is not explored.

3)	Linear Probe Assumption of Linearity
The proposed linear activation probe relies on the assumption that harmfulness signals are linearly separable within the model’s internal activation space. This simplification enhances computational efficiency but overlooks the nonlinear semantic dependencies that often define harmful or context-sensitive content.

4)	Insufficient Evaluation Scope:
The evaluation’s domain restriction to CBRN data limits the paper’s external validity. Incorporating multi-domain datasets, such as AdvBench and JailbreakBench, would demonstrate whether the proposed method generalizes beyond structured, science-based harms. A broader evaluation would substantiate the claim that the method is a general-purpose, production-ready defense rather than a domain-specific safeguard.

**Questions:**

Please refer to the weakness.

---

> ### Author Response · Authors · 2025-11-17
>
> We thank Reviewer zAes for their positive assessment and constructive feedback on our work. We've addressed the concerns raised below, and are happy to update the camera ready version of the submission in line with your feedback. We hope the clarifications made below allow you to raise your evaluation score.
>
> # W1 - Analysis of Failure Cases:
>
> > While the exchange-classifier architecture clearly improves robustness relative to previous Constitutional Classifiers, the results in Section 3 suggest that failure cases remain and deserve deeper analysis. Specifically, the system still exhibited two high-risk vulnerabilities across 226K red-teaming queries (≈ 0.00885 per thousand), implying that some jailbreaks can still bypass contextual evaluation.
>
> Thank you. We agree that failure cases remain—there are no perfectly robust machine learning systems. We'd be happy to add more information about failure cases in the Camera Ready version of the paper. In particular, with regards to the failure cases found, we found the following patterns:
> - One attack pattern is "flooding output," where attackers elicit responses containing large amounts of general information rather than specific harmful content. Because of the way the rubric grading works, these responses can sometimes include rubric keywords.
> - Prompt injection attempts insert hidden instructions within prompts to manipulate classifier behavior. While such attacks occasionally bypass exchange classifiers, they can be mitigated through improved training data curation.
> - Other attacks combine multiple techniques, like ciphers, roleplaying, other languages, sometimes with very long context.
>
> # W2 - Analysis of Failure Cases
> > While the paper provides strong empirical validation of the two-stage cascade architecture, it lacks a robustness trend on model scale, which limits the interpretability of the reported robustness gains. In Section 4, the two-stage setup combines “extra-small” and “small” classifiers, but the impact of scaling each stage independently is not explored.
>
> We thank the reviewer for this suggestion. Our work focuses on validating classifier architectures at fixed sizes and demonstrating deployment-ready performance. While we agree that systematic scaling experiments examining the independent contribution of each cascade stage would enhance interpretability, this remains important future work that we will acknowledge in the revision.
>
> # W3 - Linear Probe Assumption of Linearity:
>
> > Linear Probe Assumption of Linearity The proposed linear activation probe relies on the assumption that harmfulness signals are linearly separable within the model’s internal activation space. This simplification enhances computational efficiency but overlooks the nonlinear semantic dependencies that often define harmful or context-sensitive content.
>
> We appreciate this thoughtful concern. Our empirical results demonstrate strong performance despite the linearity assumption, and we prioritized linear probes for their computational efficiency—a critical consideration for production deployment. We have not yet evaluated nonlinear alternatives but will discuss this direction as promising future work.
>
> # W4 - Insufficient Evaluation Scope:
> > Insufficient Evaluation Scope: The evaluation’s domain restriction to CBRN data limits the paper’s external validity. Incorporating multi-domain datasets, such as AdvBench and JailbreakBench, would demonstrate whether the proposed method generalizes beyond structured, science-based harms. A broader evaluation would substantiate the claim that the method is a general-purpose, production-ready defense rather than a domain-specific safeguard.
>
> Our focus on CBRN domains is intentional and reflects the extreme stakes involved—these are precisely the domains where high classifier robustness is most critical. Notably, both OpenAI and Anthropic deploy specialized classifier safeguards specifically for CBRN content, given the severity of potential harms. Therefore, improvements to those domain-specific safeguards will be of interest to the community. We believe that demonstrating production-grade robustness in this highest-risk category, particularly with reduced computational costs, represents a valuable contribution. We will clarify this rationale in the revision and acknowledge broader domain evaluation as future work.

---

> > ### Author Response · Authors · 2025-11-21
> >
> > Dear Reviewer,
> >
> > We also wanted to let you know that we experimented with using a simple MLP probe with width 512 using our dataset of red-teaming prompts. We found that the MLP probe did NOT improve performance over the linear probe. On a different version of the dataset, both methods achieve 99.9% TPR at 0.05%FPR on WildChat. Therefore, given the probe is cheaper, we prefer it.
> >
> > Thank you.

---

### Author Response · Authors · 2025-11-17
**General Response to All Reviewers**

We thank all reviewers for their constructive feedback. We address common themes:

* Domain Focus on CBRN: Our CBRN focus aligns with Anthropic and OpenAI's prioritization for frontier model safety. CBRN represents extremely high-stakes harms where defensive failures carry catastrophic consequences. We believe demonstrating production-grade defenses for the highest-risk category is the appropriate first contribution, while broader domain evaluation remains important future work.

* Reproducibility: Due to proprietary concerns and the risk of aiding attackers, we cannot publicly release code and models. We will share datasets and evaluation methodologies with the Frontier Model Forum to enable responsible replication. In the revision, we will provide detailed architectural specifications, training procedures, and evaluation protocols.

* Presentation: We are happy to add architectural diagrams, definitions, and clearer experimental context throughout.

* New Probe Results: Since submission, we completed red-teaming of the Probe→Classifier system with ~160K red-teaming attempts and ~190 attackers.  Our results supports the claim that probes can replace first-stage classifiers and maintain sufficient robustness.

---

### Author Response · Authors · 2025-11-21
**Gentle Reminder**

Dear Reviewers,

This is a gentle reminder that we've responded to your concerns below and added new experiments. We'd love to know what you think.

Thank you.

---

### Meta-Review · Area_Chair_9bPQ · 2025-12-31

**Summary:**

Reviewers broadly agree the paper tackles an important, timely problem (robust jailbreak defenses) and makes a practically impactful contribution via exchange-level classification, a two-stage cascade for cost reduction, and activation probes, backed by unusually large-scale human red-teaming and low observed production refusal. The main points tempering enthusiasm are limited transparency/reproducibility (proprietary models/data and incomplete implementation detail), scope concerns (CBRN-centric evaluation and limited exploration of scaling/generalization), and some presentation/clarity gaps. Overall sentiment is “borderline but positive” from zAes/VZzS/HfNu, with dTVX/e8Wf below threshold largely for clarity/reproducibility, yielding a net lean toward acceptance contingent on a clearer camera-ready.

**Reviewer Concerns:**

The rebuttal substantially addressed several concrete questions: it clarified what “robustness” meant (vulnerability rate per 1k red-teaming queries) in response to dTVX, clarified that the evaluated exchange-classifier system did not also use an output classifier and explained the 569K vs 226K query discrepancy as separate red-teaming campaigns for e8Wf, and answered calibration/softmax-loss questions for VZzS. Importantly, it also strengthened the probe story by adding new adversarial validation (addressing HfNu’s key concern) and by reporting that a simple MLP probe did not outperform the linear probe. Outstanding concerns remain around (i) reproducibility and independent verifiability given proprietary models/data (raised most strongly by e8Wf and dTVX), (ii) limited cross-domain benchmarking beyond CBRN and limited scaling/ablation depth (noted by zAes/VZzS), and (iii) the need for clearer exposition (system diagram, tighter definitions, and better contextualization vs standard I/O filtering) emphasized by dTVX/e8Wf.

**Reviewer Scores:**

zAes: likely unchanged, as rebuttal mostly reframed limitations as future work without new evidence on broader generalization.
VZzS: already indicated an increase after rebuttal; net effect likely modest given most questions were answered but core reproducibility/domain-scope issues persist.
dTVX: could move slightly upward if the camera-ready delivers substantial clarity (diagram, clearer baselines/metrics, stronger positioning vs I/O filtering), but given the continued proprietary constraint I’d expect at most a small bump.
e8Wf: explicitly stated they will keep the score (≈ +0).
HfNu: explicitly increased after the new probe red-teaming evidence; net effect likely a clear bump).

---

### Decision · Program_Chairs · 2026-01-26

Accept (Poster)